# Birefringence-induced phase delay enables Brillouin mechanical imaging in turbid media

Giuseppe Antonacci [1] ✉, Renzo Vanna [2], Marco Ventura[2], Maria Lucia Schiavone [3], Cristina Sobacchi[3,4], Morteza Behrouzitabar[5], Dario Polli [1,2,5], Cristian Manzoni [2] ✉ & Giulio Cerullo [2,5]

Acoustic vibrations of matter convey fundamental viscoelastic information that can be optically retrieved by hyperfine spectral analysis of the inelastic Brillouin scattered light. Increasing evidence of the central role of the viscoelastic properties in biological processes has stimulated the rise of non-contact Brillouin microscopy, yet this method faces challenges in turbid samples due to overwhelming elastic background light. Here, we introduce a common-path Birefringence-Induced Phase Delay (BIPD) filter to disentangle the polarization states of the Brillouin and Rayleigh signals, enabling the rejection of the background light using a polarizer. We demonstrate a 65 dB extinction ratio in a single optical pass collecting Brillouin spectra in extremely scattering environments and across highly reflective interfaces. We further employ the BIPD filter to image bone tissues from a mouse model of osteopetrosis, highlighting altered biomechanical properties compared to the healthy control. Results herald new opportunities in mechanobiology where turbid biological samples remain poorly characterized.

Mechanical properties regulate functional biological processes across different temporal and spatial scales. Disruption of such processes caused by altered biomechanics is closely associated with the development and progression of severe diseases. In cancer, for example, malignant cells exhibit reduced stiffness, promoting their invasiveness through increased deformability[1,2]. Similar implications arising from altered viscoelastic properties are observed in eye[3], neurodegenerative[4], and bone[5] disorders. As a result, gaining a comprehensive understanding of these alterations is crucial for improved diagnosis and treatment.

Unfortunately, conventional elastography methods that measure these properties require physical contact with the sample, making them unsuitable for in vivo and three-dimensional environments. Consequently, there is a growing demand in biomedicine for non-

invasive optical elastography methods to assess the biomechanical properties[6]. To answer this need, Brillouin microscopy has emerged as a purely optical and label-free method to measure viscoelastic properties with sub-cellular spatial resolution in the volume of biological organisms[7].

In Brillouin spectroscopy, light exchanges energy with spontaneous acoustic vibrations of matter (or *acoustic phonons*) whose propagation velocity and lifetime depend on the viscoelastic properties of the material. As a consequence of the energy exchange, a portion (~$10^{-12}$) of the incoming light exhibits a frequency shift defined by $\nu_B = \pm(2n/\lambda)V\sin(\theta/2)$, where $n$ is the refractive index of the medium, $\lambda$ is the wavelength of the probe laser, $V$ is the acoustic velocity inside the medium and $\theta = \cos^{-1}(\hat{\mathbf{k}}_{in} \cdot \hat{\mathbf{k}}_s)$ the scattering angle defined by the incident and scattered unit wave vectors $\hat{\mathbf{k}}_{in}$ and $\hat{\mathbf{k}}_s$. This inelastic

[1]Specto Photonics, Via Giulio e Corrado Venini 18, 20127 Milano, Italy. [2]CNR-Istituto di Fotonica e Nanotecnologie, CNR-IFN, Piazza Leonardo da Vinci 32, 20133 Milano, Italy. [3]IRCCS Humanitas Research Hospital, via Manzoni 56, 20089 Rozzano (Milano), Italy. [4]CNR-Istituto di Ricerca Genetica e Biomedica (CNR-IRGB), UOS di Milano, via Fantoli 16/15, 20138 Milano, Italy. [5]Dipartimento di Fisica, Politecnico di Milano, Piazza Leonardo da Vinci 32, 20133 Milano, Italy. ✉e-mail: giuseppe@spectophotonics.com; cristianangelo.manzoni@cnr.it

process gives rise to a Stokes and an anti-Stokes Brillouin spectral peak that are typically shifted by a few GHz from the elastic Rayleigh peak. Detection of the Brillouin peaks and the measurement of the associated frequency shift and linewidth provides crucial information about the viscoelastic properties of samples, and in particular on their specific elastic moduli that form the full elastic tensor[8]. In the back-scattering geometry, where $\theta = \pi$, the commonly measured quantity is referred to as the complex longitudinal modulus $\mathbf{M} = M' + iM''$, where $M' = V^2/\rho$ is the storage modulus associated to the material compressibility with $\rho$ being the material density, and $M'' = 2\pi\nu_B\eta$ is the loss modulus associated to the longitudinal viscosity $\eta$. While spectral broadening is minimized in such configuration[9], the ultimate spatial resolution of the technique must also take into account the extension of the probed acoustic phonons that may be larger than the system Point Spread Function (PSF)[10].

The non-destructive and label-free nature of Brillouin spectroscopy has motivated its adoption as a novel 3D imaging tool for mechanobiology, enabling the biomechanical characterization of sub-cellular structures within living cells[11–13] and their changes in response to external stimuli or intracellular liquid-to-solid phase transitions[14,15]. Recently, Brillouin microscopy has been further turned into a novel clinical instrument for ophthalmology[16,17] and holds great promise to become a valuable tool for cardiovascular event prevention[18] and cancer disease diagnosis[19,20].

Despite the growing demand amongst the biomedical[21] and material science[22] communities, Brillouin microscopy still faces fundamental barriers in the adoption as a consequence of the challenge in detecting the weak Brillouin peaks overlapped with the strong elastic background light[23,24]. Recent advancements have shown significant improvement in image acquisition speed, both in stimulated[25–27] and spontaneous[28,29] scattering regimes. However, the low number and tight frequency shift of inelastically scattered photons still impose a fundamental obstacle when assessing opaque and turbid biological samples. This obstacle is a consequence of the dominant elastic background light originating from Rayleigh scattering and specular Fresnel reflections, whose intensity can easily be more than a million times stronger than the Brillouin scattered light in non-transparent samples. As a result, detection of the Brillouin peak requires the use of spectrometers with high spectral contrast, defined as the transmission peak-to-background ratio, or the employment of ultra-narrowband filters.

Tandem multi-pass Fabry–Pérot interferometers have been traditionally used and still remain widely adopted in Brillouin spectroscopy. These devices offer unparalleled spectral contrast (~150 dB) and resolution (~100 MHz). However, their application within the life sciences is limited by the relatively slow scanning process of the cavity imposed by the high-parallelism requirement for the mirrors, resulting in data acquisition times of >0.5 s per spectrum[30]. The recent introduction of virtually-imaged-phased-arrays (VIPAs)[31], a modified type of Fabry–Pérot etalon, has significantly reduced the acquisition time down to ~100 ms, in turn enabling the extension of Brillouin spectroscopy from a point-sampling technique into a novel all-optical mechanical imaging method[32]. The speed enhancement is a consequence of the scan-free parallel detection of the Brillouin spectrum that is generated by the VIPA's interferometric pattern in the spatial domain. On the other hand, VIPA etalons are intrinsically limited by a low spectral contrast of ~30 dB, enabling the detection of mainly transparent samples, such as liquids, in specific dark-field configurations[33].

Several methods have been successfully demonstrated to increase the detection capabilities of modern spectrometers, either by increasing the spectral contrast or by background removal. Cascading of two crossed VIPAs is one of the most widely adopted methods[34], providing a sufficient (~60 dB) spectral contrast to probe semi-transparent tissues and cells. The use of a Lyot filter at the Fourier plane of the VIPA has been demonstrated as a useful strategy to remove the high-frequency content of the spectrum, resulting in an extra 20 dB increase[35]. Integration of rhomboidal diffraction masks has also been demonstrated to be a simple yet effective solution to deflect the elastic background light away from the dispersion axis[14], while field apodization has been introduced in an attempt to convert the exponentially decaying output beam of the VIPA into a Gaussian-like beam, in turn resulting in a contrast enhancement of 20–30 dB depending on the equalization mask used[11,36]. Alternative approaches to reject the elastic background light involve interferometric schemes based on e.g., Michelson[37] or Mach-Zehnder[38] interferometers aiming at a destructive interference of the elastic background light. A similar principle has been proposed using etalon-based notch filters providing a remarkable 40 dB extinction in a 4-pass configuration[39,40].

While the aforementioned methods have greatly contributed to extending the field of application of Brillouin microscopy within the Life Sciences, these involve several optical elements and typically multiple optical paths that affect the system robustness, stability and throughput. One of the most commonly employed methods that has recently enabled the emergence of high-speed line-scan and stimulated Brillouin microscopy involves the use of Rubidium gas cells that exhibit sharp absorption lines[41] at the laser frequency. This approach can provide excellent suppression capability of nearly ~50 dB by heating the gas at a temperature around 100 °C, yet it requires stabilization of the laser emission wavelength and introduces additional attenuation and asymmetry to the Brillouin peaks as a consequence of the multiple narrow absorption lines across the spectrum. Moreover, this method currently works only for specific wavelengths and lacks suitable molecular lines that can allow operation at 660 nm, a widely adopted wavelength in Brillouin microscopy as it offers a trade-off between spatial resolution, signal strength (scaling as $\lambda^{-4}$) and sample phototoxicity[42]. More recently, an on-chip notch filter based on a silicon nitride ring resonator with 10 dB extinction ratio has been demonstrated as a method to reduce the system complexity and footprint[43], yet further work is needed to make this a viable solution.

Here we introduce a Birefringence-Induced Phase Delay (BIPD) filter that is capable of effectively rejecting the elastic background light with an unprecedented extinction ratio of 65 dB. Unlike previous approaches, the BIPD filter is common-path and can simultaneously operate at all visible and near infrared wavelengths. Integrating the BIPD filter as a compact, fiber-coupled module prior to the spectral detection system consisting of a single-stage and single-pass VIPA spectrometer, we demonstrate its rejection capability by acquiring Brillouin spectra in 3D environments of extreme turbidity and in imaging configurations of high reflectivity, where the overwhelming elastic background light would otherwise hinder the detection of the Brillouin peaks. Recording Brillouin images of highly scattering murine vertebra bone tissues, we found significantly altered biomechanical properties in the Rankl knockout mouse as a model of human infantile malignant osteopetrosis[44,45], a rare genetic disorder characterized by a generalized increase in bone density owing to lack of bone resorption. Results pave the way to a wider application in diseases affecting highly opaque tissues, such as bone and dentin, that previously required the use of slow scanning multi-pass Fabry–Pérot interferometers for Brillouin mechanical assessment[46–48].

## Results

### High-rejection BIPD filter

Figure 1a illustrates the general concept of our common-path BIPD filter. When monochromatic light travels through an anisotropic birefringent crystal with optical axis oriented in the plane perpendicular to the propagation direction and aligned at 45° with respect to the input polarization, the ordinary and extraordinary components of the field experience different refractive indexes $n_o$ and $n_e$, respectively, so that after propagation through a distance $L$ a phase retardation

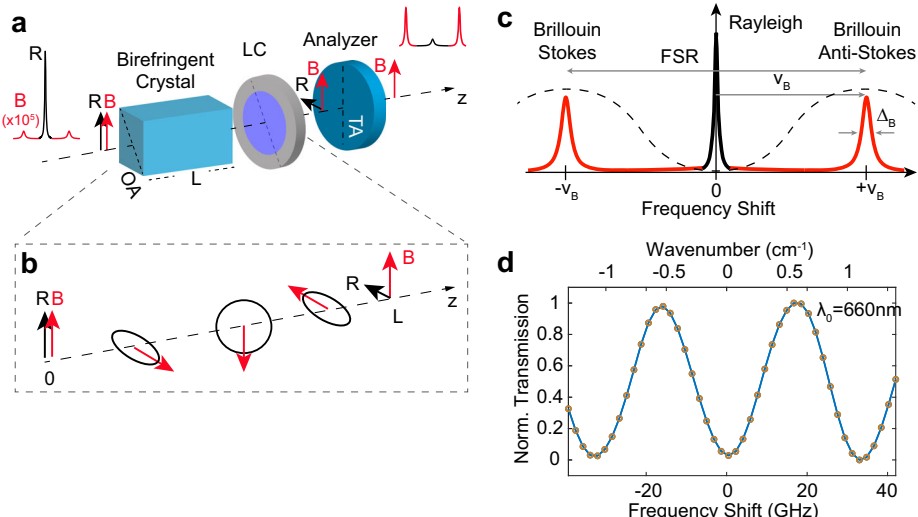

**Fig. 1 | BIPD operation principle. a** Filter schematic. The Rayleigh and Brillouin signals have equal linear polarization states and enter a birefringent crystal with its optical axis (OA) perpendicular to $\hat{k}$ and aligned at $45°$ with respect to the input polarization. **b** Phase delay along the birefringent media. The anisotropy of the birefringent crystal induces a phase delay so that after a specific length L the output signals exhibit a relative phase retardation $\Delta\phi = \pi$. A full-wave liquid crystal (LC) retarder ensures linear and orthogonal polarization states for the Rayleigh and Brillouin signals by fine tuning the phase delay of the beam transmitted by the birefringent crystal. An analyzer is used to suppress the elastic background light while transmitting the disentangled Brillouin scattered light by an appropriate orientation of its transmission axis (TA). **c** Sinusoidal spectral transmission function (dashed line) of the filter along with a representative Brillouin spectrum. The filter bandwidth is set by an appropriate choice of the FSR to maximize the transmission of the Brillouin peaks (here magnified by >$10^5$ for easier visualization). **d** Experimental characterization of the filter transmission function in a normalized intensity scale. The transmission was measured by sending a broadband incoherent light through the filter, and measuring the transmitted light by a FTIR spectrometer with high spectral resolution. The experimental FSR for $\lambda_0 = 660$ nm central wavelength is measured to be $(33.0 \pm 0.5)$ GHz (1.1 cm$^{-1}$).

$\phi = (2\pi/\lambda)L\Delta n$, where $\Delta n = |n_e - n_o|$, is accumulated between them. This corresponds to a change in the field polarization state. We assume that the excitation has linear polarization, and that the Brillouin scattered light has equal polarization state to that of the elastic background (Rayleigh) light. Since the phase delay $\phi$ introduced by the birefringent crystal is wavelength dependent, it causes a relative variation between the Brillouin and Rayleigh polarization states (Fig. 1b). For a given material birefringence $\Delta n$, it is therefore possible to select a crystal thickness such that the relative phase variation $\Delta\phi = \pi$. This condition occurs for a crystal length equal to

$$L = \frac{c}{2\nu_B}\frac{1}{\Delta n}, \qquad (1)$$

where $c$ is the speed of light (Supplementary Note 1). This relationship is equivalent to posing that the Free Spectral Range (FSR) of the filter is equal to FSR $= c/L\Delta n$. In addition, a fast full-wave liquid crystal (LC) retarder enables active fine-tuning of the phase retardation of the beams transmitted by the birefringent crystal to guarantee a linear polarization state for the Rayleigh signal. As a result, the Rayleigh signal is fully rejected by a linear polarizer (analyzer) with transmission axis oriented orthogonal to its polarization. Conversely, the Brillouin signal with polarization orthogonal with respect to the Rayleigh signal and parallel to the analyzer transmission axis is transmitted (Fig. 1c).

In our experimental setup we chose Yttrium Orthovanadate (YVO$_4$) as the birefringent material that, compared to other uniaxial crystals, provides high birefringence ($\Delta n > 0.2$) and transmission (>80%) for a broad spectral window in the visible and near-infrared (NIR) regions (see Methods). A high birefringence is especially convenient to avoid unpractically long crystals. We experimentally characterized the transmission function and the associated FSR for the crystal length $L = 35$ mm by coupling a broadband light source to the BIPD filter and acquiring the transmitted signal using a FTIR spectrometer (Jasco 6800) with high (2.1 GHz) spectral resolution. Experimental data show the expected sinusoidal transmission function with a measured FSR ~ $(33.0 \pm 0.5)$ GHz (Fig. 1d). This value reveals that our YVO$_4$ crystal has an effective birefringence of $\Delta n$ ~ $0.26 \pm 0.02$, in agreement with previously reported values[49]. An extinction ratio of up to $(64.8 \pm 1.0)$ dB was measured coupling the monochromatic light of a single-longitudinal-mode laser to the filter and recording the resulting maximum and minimum transmitted powers by fine tuning the LC retarder, and after scaling with respect to attenuation levels of calibrated neutral density filters used to overcome the limited dynamic range of commercial photodetectors. The same optical configuration was used to characterize the optical loss of the filter in transmission, which was measured to be $0.4 \pm 0.1$ dB at the transmission maximum. Unlike traditional filtering schemes that are commonly adopted in Brillouin microscopy, we further demonstrate the possibility to apply the BIPD filter to a wide spectral range of visible and NIR wavelengths (Supplementary Fig. 1).

## BIPD-assisted Brillouin microscope

A custom confocal Brillouin microscope was built to demonstrate the full capability of the filter (Fig. 2a, Methods). The microscope included a single-stage, single-pass VIPA spectrometer module with measured spectral contrast of $30 \pm 3$ dB (Supplementary Fig. 2), spectral resolution of $515 \pm 20$ MHz and sensitivity of $14.5 \pm 2$ MHz (Supplementary Fig. 3). Combining the individual contributions of the BIPD filter and the VIPA, we estimated a total system dynamic range of >90 dB. A first proof-of-concept of the filter suppression capability was obtained in controlled setting using pure distilled water as a test sample placed inside a cell culture dish with a glass substrate. The laser beam was focused inside the dish, ensuring that the focal spot was axially distanced by >100 μm from the glass substrate to minimize the collection of specular reflections. Despite the relaxed settings in terms of both sample transparency and optical arrangement, the Brillouin peaks could not be observed when the transmission minimum of the BIPD filter was slightly misaligned with respect to the laser wavelength (Fig. 2b). In fact, a minimal yet unavoidable amount of beam reflections arising along the optical path of the microscope was sufficient to

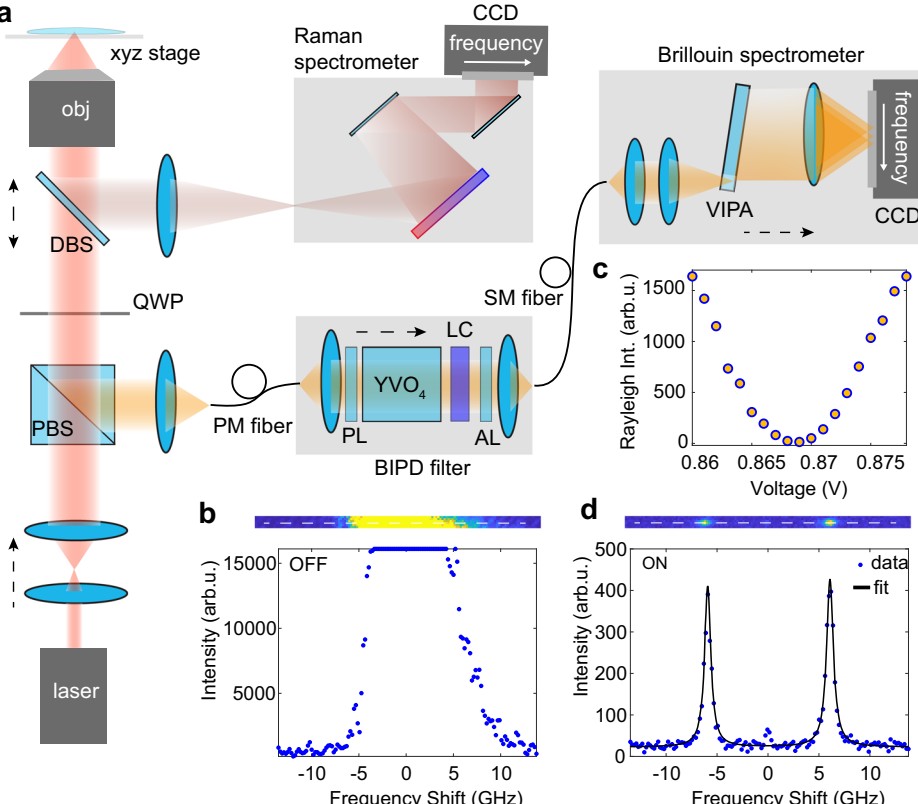

**Fig. 2 | Background suppression by BIPD filtering. a** Schematic of the custom confocal Brillouin microscope. A dichroic beam splitter (DBS) reflects light at wavelengths longer than $\lambda = 670$ nm for Raman spectral analysis while transmitting the Rayleigh and Brillouin scattered light to a polarizing beam splitter (PBS). A polarization-maintaining single-mode (PM) fiber gives confocality, flexible beam delivery and controlled polarization direction to the filter and detection units. The filter module comprises a fiber collimator, a polarizer (PL) with the transmission axis parallel to the input polarization and the YVO₄ crystal with the optical axis perpendicular to the propagation direction and rotated by 45° with respect to the input polarization. The full-wave LC retarder actively tunes the phase delay of the beam transmitted by the YVO₄ crystal, ensuring a perpendicular linear polarization for the Rayleigh signal with respect to the transmission axis of the analyzer (AL). A single-mode (SM) fiber delivers the background-free Brillouin signal to the spectrometer, which comprises a fiber collimator and a cylindrical lens to focus light to a single VIPA etalon. A second cylindrical lens yields the final spectrum by Fourier transform for acquisition by a CCD camera. **b** Brillouin spectrum of distilled water. With the filter transmission minimum detuned from the laser wavelength, the Brillouin peaks are overwhelmed by the residual and unavoidable amount of specular reflection arising along the microscope optical path. By applying a voltage to the LC retarder (**c**), the filter can be tuned to suppress the Rayleigh peak, making the Brillouin peaks clearly visible (**d**). Error bars are given by the fitting error.

overwhelm the Brillouin spectrum and saturate the CCD camera. By actively tuning the LC retarder (Fig. 2c), we aligned the transmission minimum of the filter to the laser wavelength. In turn, the Rayleigh peak was completely suppressed, making the Brillouin peaks clearly visible (Fig. 2d and Supplementary Movie 1). Moving the transmission maximum of the VIPA between two consecutive interference orders, we further characterized the filter transmission function, providing additional evidence of the measured FSR (Supplementary Fig. 4). Moreover, we observed no significant variations in the Brillouin peak shape and frequency shift in response to either a varying phase delay (Supplementary Fig. 5) or a change in the output polarization of the Rayleigh and Brillouin signals that may be caused by small sample birefringence (Supplementary Fig. 6).

Small phase variations arising from thermal expansion of the YVO₄ crystal (typically in the order of $2.2 \times 10^{-6} K^{-1}$)[50] may affect the stability of the filter for environmental thermal fluctuations of >0.01 °C. Moreover, an extinction level above 60 dB further imposes a laser stability of <10 MHz, which is a non-trivial requirement and typically requires a laser locking. To compensate for extra phase delays induced by unavoidable environmental temperature fluctuations (Supplementary Fig. 7) and/or frequency drifts of the laser source - occurring on a timescale of minutes in our system (Supplementary Fig. 8) - we implemented a closed-loop control tuning the voltage applied to the LC retarder and using the resulting Rayleigh intensity as

feedback, in turn ensuring high stability over long (>60 min) data acquisition (Supplementary Fig. 9).

## Suppression of Rayleigh scattering in turbid media

To demonstrate the capability of the filter to acquire Brillouin spectra of highly turbid media, we conducted measurements of milk:water solutions at different concentrations and depths. Results showed that the Rayleigh peak remained negligible up to 1:1000 concentrations, for which the amount of elastically scattered light was already orders of magnitude higher compared to the Brillouin scattered light (Fig. 3a, b). The measured Brillouin spectra exhibited a higher frequency shift (Fig. 3c) and a broader linewidth (Fig. 3d) compared to distilled water as the milk concentration increased. Despite the increasing Rayleigh peak at critical concentrations above 1:100, the Brillouin peaks were still visible even in the extreme case of 100% whole milk, where the spectrum exhibited peaks of lower intensity as a direct consequence of the strong elastic scattering (Fig. 3e). Focusing the illumination beam inside the whole milk, we observed Brillouin peaks up to a depth of $z = 350 \pm 5$ µm corresponding to an optical density of OD = 7 (Supplementary Fig. 10). Notably, despite a small decrease in the frequency shift along $\hat{z}$, the Brillouin peaks did not exhibit visible shoulders as a consequence of multiple scattering (MS)[51]. This absence may be linked to the fact that, in the MS regime, light becomes randomly polarized compared to the incident one. As a result, the input polarizer of the

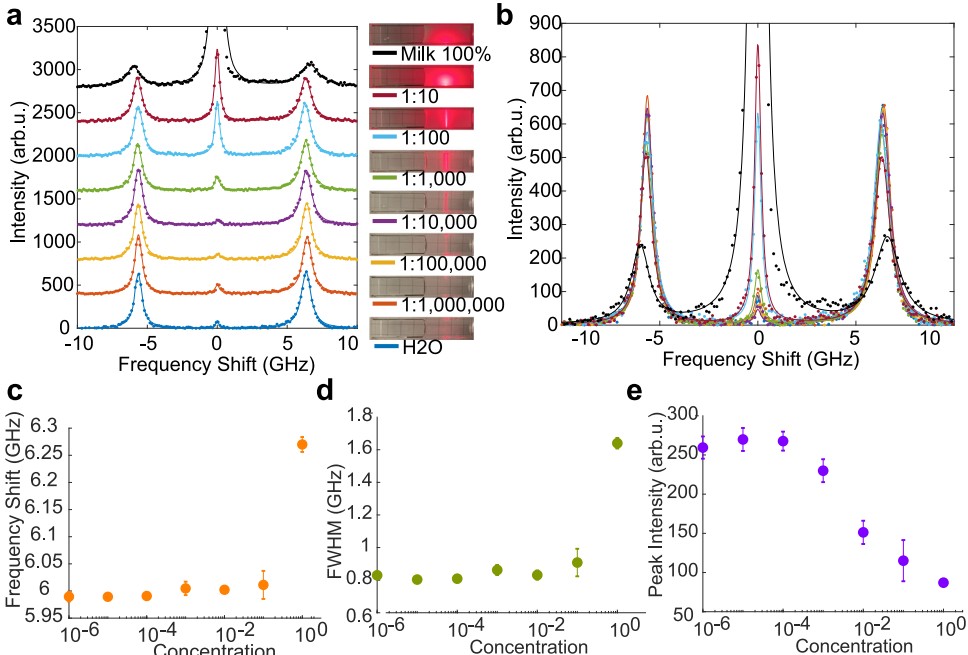

**Fig. 3 | BIPD filter enables acquisition of Brillouin spectra in turbid media.**
**a**, **b** Acquired spectra of water-milk solutions at different concentrations. Light diffused by elastic scattering increases with whole milk concentration (inset). Such increase is not reflected in the associated Brillouin spectra where the Rayleigh peak was fully rejected by the filter even up to high (1:1000) concentrations. The same data is visualized with (**a**) and without (**b**) the addition of a constant baseline

between the spectra. Both frequency shift (**c**) and linewidth (**d**) increased with milk concentration as a result of the different viscoelastic properties of milk as well as the contribution from multiple scattering. Despite the lower intensity caused by the natural attenuation of the laser beam propagating through a highly scattering medium (**e**), the Brillouin peaks remain clearly visible also in pure milk.

BIPD filter rejects half of the MS light signal, thereby reducing the presence of the expected shoulders.

## Rejection of specular reflections

While elastic scattering is the dominant process in turbid media, specular Fresnel reflections arising along the optical path significantly contribute to the formation of strong Rayleigh peaks. In standard confocal imaging settings, for example, the optical sectioning is typically performed in the vicinity of the glass cover slip, where cells apply adhesion forces[52]. In these settings, a high amount of specular reflection is collected by the system, making Brillouin imaging of cells particularly challenging without the aid of specific dark-field configurations[32,33] or extra thin layers that are spin-coated on top of the glass cover slip to smooth the refractive index contrast at the sample interface[11]. We validated the system resilience against specular reflections by scanning the illumination beam across a standard glass-water interface in the axial direction and detecting the resulting spectrum (Fig. 4a). When the beam focus was located inside the cover slip (Fig. 4b), only the associated Brillouin peaks of silica were visible, with the Rayleigh peak being fully suppressed. On the other hand, when the beam focus was translated axially, both peaks of glass and water could be clearly observed, leaving the Rayleigh peak below the intensity threshold even in extreme conditions where the amount of specular Fresnel reflections was maximum. Measurement of the Brillouin peak intensity across the interface enabled the characterization of the system edge spread function (ESF, Fig. 4c), whose first derivative enabled the retrieval of the PSF (Fig. 4d) of the Brillouin microscope[53] showing an axial resolution of $(7.8 \pm 0.2)\mu m$.

## Brillouin imaging of osteopetrotic bone

The BIPD filter was finally demonstrated to enable Brillouin mechanical imaging on highly scattering and turbid bone tissues that were previously inaccessible using single-stage and single-pass VIPA spectrometers. The imaging system was first tested under controlled settings

by acquiring Brillouin maps of both frequency shift and linewidth of a mixture of poly-methyl-methacrylate (PMMA) and polystyrene (PS) beads of ~8 μm and ~10 μm diameter respectively, on a fused silica coverslip (see Methods). The resulting image contrast proved the microscope capability to spatially resolve and spectrally differentiate beads of different viscoelastic properties (Supplementary Fig. 11). Next, we addressed the capability of the microscope to assess the micromechanical properties of unstained histological sections of vertebra from both healthy (Fig. 5a–c) and osteopetrotic Rankl knockout (Fig. 5d–f) mice. Despite the extreme turbidity of the sample analyzed (Supplementary Fig. 12) involving high Rayleigh scattering and thus a substantially attenuated signal strength compared to typical semitransparent samples, the BIPD-assisted Brillouin microscope was able to acquire Brillouin frequency and linewidth maps over large areas with 300 ms pixel dwell time. Remarkably, this represents a significant decrease in the data acquisition time with respect to previously reported studies on similar samples involving a 6-pass scanning Fabry–Pèrot interferometer as the detection unit[46,47]. A general overview of the acquired Brillouin maps reveals trabecular (spongy) bone structures associated with relatively high Brillouin shift and linewidth, in agreement with the mineral structure and content of bone. In parallel, the bone marrow, containing hematopoietic and stromal cells filling the space between the trabeculae, is associated with relatively lower Brillouin shift and linewidth. In order to specifically analyze the biomechanical properties of bone structures only, Raman spectra and Brillouin maps were sequentially registered (Fig. 5g–h) aiming to analyze Brillouin data associated with pixels containing mineralized bone only, thanks to segmentation of Raman data containing apatite signals (Supplementary Fig. 13). The extracted bone data subset was found in the high-frequency tail of both shift and linewidth distributions of the full spectral image data (Fig. 5i–l). This was in line with the expectation of the bone as the hardest component in the analysed tissue sections comprising also bone marrow and muscle. Notably, we found remarkably altered biomechanical properties on the retrieved

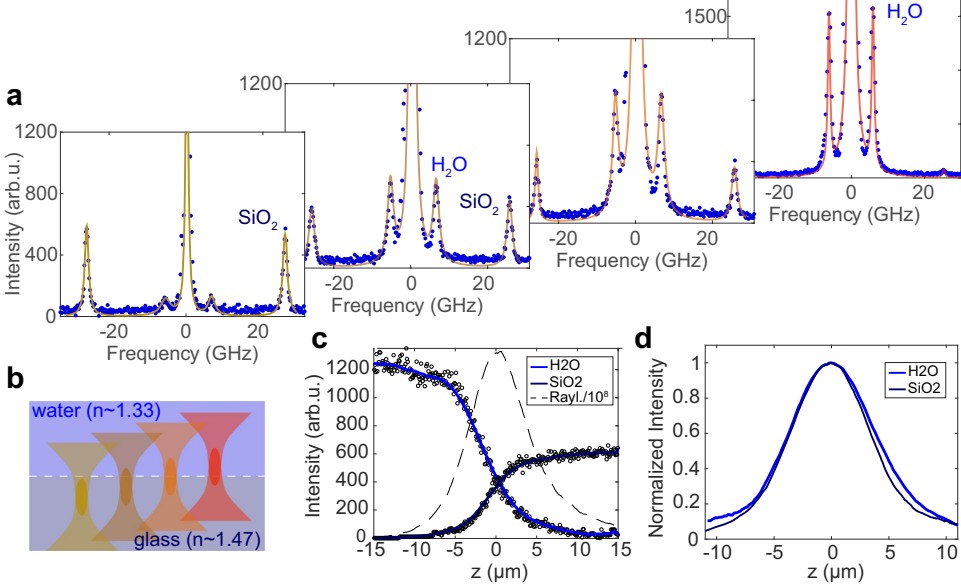

**Fig. 4 | Rejection of specular reflections. a** Brillouin spectra along a glass-water interface acquired by shifting the objective lens in the axial direction with step size of 2 μm. With the beam focus located inside the glass cover slip, only the Brillouin peaks of silica were visible with a measured frequency shift of $\nu_B = 25.8 \pm 0.1$ GHz. Brillouin peaks of water appear when the focus passes through the glass-water interface where specular reflections are dominant against Rayleigh scattering (**b**).

The Brillouin peaks remain clearly resolved with the Rayleigh peak at low levels even at depths where specular Fresnel reflections are predominant. **c** ESF obtained measuring the Brillouin peak intensity of silica and water along their interface. Dashed line represents the measured Rayleigh intensity scaled by a factor of $10^8$. **d** The PSF retrieved from the first derivative of the ESF provided an axial resolution of $(7.8 \pm 0.2)$μm.

bone data set (Fig. 5m, n). Specifically, the bone of the osteopetrotic Rankl knockout mice showed a significantly higher ($p < 0.001$) frequency shift ($\nu_B^{Rankl} = 10.74$ GHz) and linewidth ($\Delta\nu_B^{Rankl} = 5.93$ GHz) compared to the healthy control ($\nu_B^{HC} = 10.42$ GHz; $\Delta\nu_B^{HC} = 5.21$ GHz), suggesting that the osteopetrotic bone is more rigid in accordance with the higher bone content across the sample volume. This peculiar feature might contribute to a generally reduced bone quality and to increased fragility of the osteopetrotic bone.

## Discussion

In this work, we highlight bone material features routinely assessed with destructive approaches by applying a purely optical, label-free method. Moreover, the direct investigation of the micromechanical properties of bone adds unprecedented insights into the causes of reduced bone quality and enhanced fragility leading to spontaneous fracture in the osteopetrotic patients. In particular, to the best of our knowledge this is the first attempt to characterize the biomechanical properties of bone from a lethal form of osteopetrosis, at variance with reports in literature dealing with benign forms of the disease[54–57]. Similar implications can also be found in other severe and more common diseases such as osteogenesis imperfecta and osteoporosis[58,59]. However, characterization of bone tissues has remained an open challenge in Brillouin microscopy as a consequence of the high Rayleigh scattering and, at the same time, extremely weak Brillouin signals due to the limited propagation depth. In this regard, the osteopetrotic bone has represented an even greater challenge, owing to its extremely high density. Our BIPD filter addresses this challenge, enabling the all-optical biomechanical assessment of bone samples using scanless single-stage VIPA spectrometers.

While existing solutions involve multi-stage and multi-path detection schemes, our BIPD filter is common-path and exhibits an unprecedented extinction ratio of ~65 dB in a single optical pass. These unique features make the filter ultra-compact with an overall size of 5 × 15 cm (Supplementary Fig. 14), providing high stability against mechanical drifts. Long-term stability against environmental temperature fluctuations was achieved through a closed-loop control

correcting for the extra phase retardation induced by thermal expansion of the crystal. Notably, the theoretical suppression capability of the BIPD filter is ultimately dictated by the quality of the employed polarizer and analyzer, which in the present study have a measured extinction ratio of >90 dB. In practice, however, such performance is not trivial to achieve as a consequence of the limited spectral purity of the laser, the tight (<1 mrad) tolerance in the analyzer transmission angle as well as the intrinsic crystal impurities that slightly diffuse the incoming light and induce a small amount of random phase photons. Exploiting the phase modulation induced by a birefringent crystal to disentangle the relative polarization states of the Brillouin and Rayleigh scattered light signals, our filter can simultaneously operate at all visible and NIR wavelengths and provides high compatibility with the existing Brillouin microscopes.

The successful demonstration of the BIPD filter enables all-optical mechanical measurements in tissues characterized by high turbidity, such as bone and dentin, that were previously inaccessible using scanless single-stage spectrometers. Our results remove a significant barrier in the adoption of Brillouin microscopy for critical applications where raw and unprocessed biological samples need to be mechanically investigated, promoting new avenues for biomedical research and clinical applications.

## Methods

### BIPD filter module

The birefringent $YVO_4$ crystal is mounted between two Glan-Taylor polarizers (DGL10 Thorlabs; measured extinction of 90.7 dB when their transmission axes are crossed) with parallel and horizontal transmission axes (Fig. 1a). While the output polarizer (analyzer) is needed to reject the Rayleigh component, the input polarizer is convenient to increase the degree of polarization and to finely balance the ordinary and extraordinary components by rotating the input polarization with respect to the optical axis of the $YVO_4$ crystal. The latter is on a plane parallel to the input/output surfaces, and is rotated by 45° with respect to the transmission axes of the input and output polarizers. The input/output surfaces of the crystal have anti-reflection

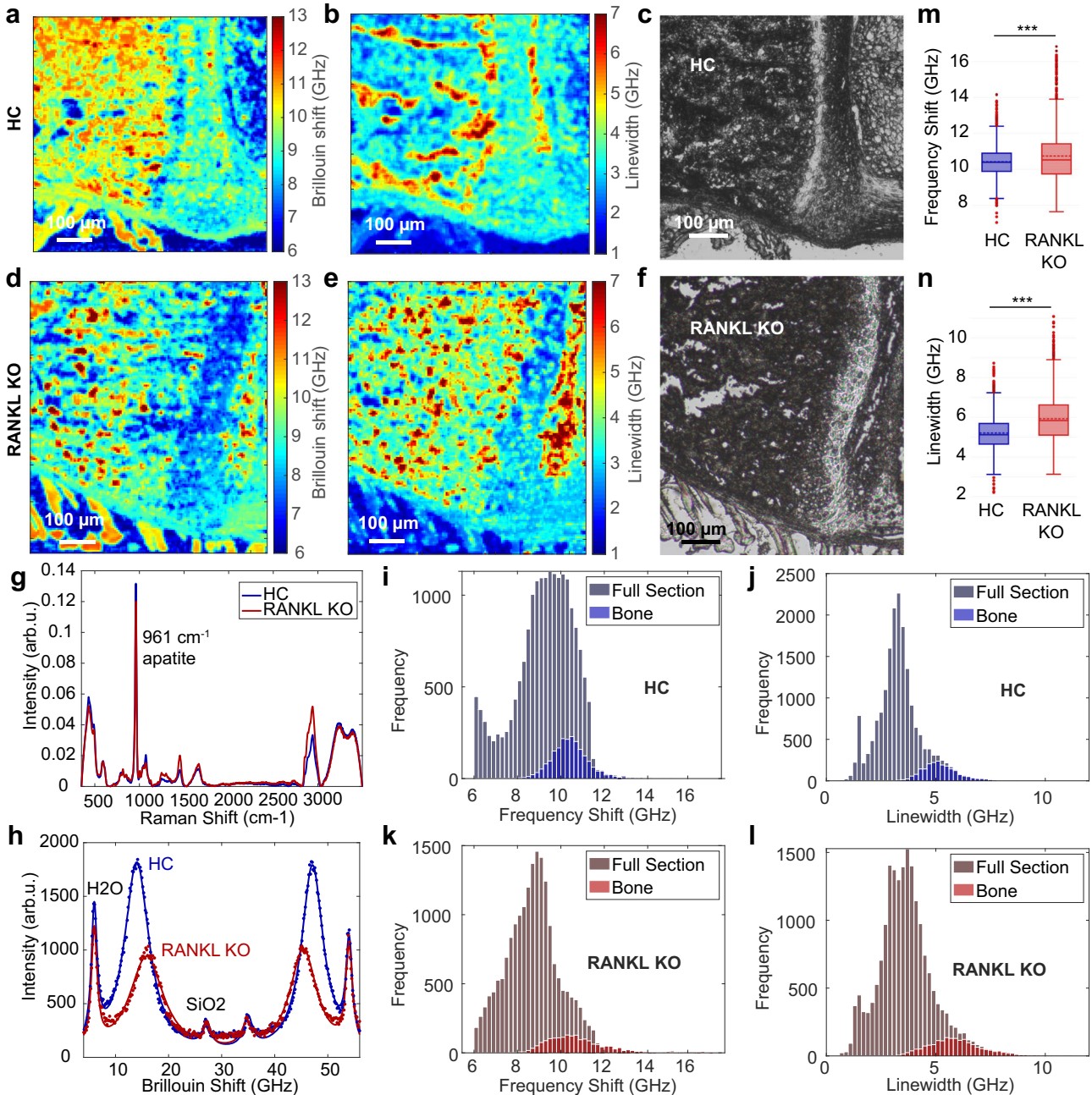

**Fig. 5 | Altered biomechanical properties in osteopetrotic mice. a–f** Brillouin frequency shift and linewidth maps, and associated bright-field images acquired on unstained histological vertebrae sections of healthy control (HC, **a–c**) and osteopetrotic Rankl knockout (**d–f**) mice, respectively. Representative Raman spectra of mineralized bone showing strong apatite signals at 961 cm$^{-1}$ (**g**) and representative associated Brillouin spectra sequentially registered at the same location (**h**). Histograms of the recorded Brillouin frequency shift and linewidth maps of the HC (**i, j**) and Rankl knockout (**k, l**) mice. The mineralized bone content distribution was retrieved through an image segmentation with the associated Raman maps identifying only pixels ($N_{HC}$ = 2017; $N_{KO}$ = 2201) reporting apatite signals. Regions containing bony mineralized structures exhibited higher Brillouin frequency shift and linewidth compared to the other soft tissue components such as bone marrow cells and connective tissue. Whisker plot of frequency shift (**m**) and linewidth (**n**) values limited to the mineralized bone regions. Both frequency shift and linewidth of bone were found to be significantly higher (***$p$ < 0.001) in Rankl knockout mice ($\nu_B^{Rankl}$ = 10.74 GHz; $\Delta\nu_B^{Rankl}$ = 5.93 GHz) compared to those in HC ($\nu_B^{HC}$ = 10.42 GHz; $\Delta\nu_B^{HC}$ = 5.21 GHz), suggesting an increased stiffness and viscosity for the osteopetrotic bone. Horizontal dashed and full lines indicate mean and median values respectively.

coatings, and a clear aperture of $10 \times 10$ mm$^2$. A general good practice for the choice of the crystal length is to obtain a FSR that matches at least half of the FSR imposed by the selected VIPA. In the present study, the crystal length was chosen to be $L = 35$ mm, providing a measured FSR = 33 GHz. A full-wave phase retarder (Thorlabs, LCC1613-A) is used to fine tune the phase delay at the output of the YVO$_4$ crystal. The phase retarder employs a nematic liquid crystal cell acting as a variable wave plate, allowing active control over the phase delay of the

incoming beam by an applied voltage. Specifically, the liquid crystal cell can induce a variable phase retardation within the range of (0–1.2) $\lambda$ with a typical response time of <30 ms at −20 to 40 °C nominal operating temperature. As a result, this active element is needed to rapidly set the Rayleigh scattering to be linearly polarized along the orthogonal direction with respect to the transmission axis of the output analyzer, thus correcting for possible laser drifts or thermal expansion of the crystal. To avoid optical losses, the filter is optically

coupled to the output of a polarization-maintaining single-mode optical fiber (Thorlabs P3-488PM-FC-2), in which its slow axis is set parallel to the transmission direction of the first polarizer. A single-mode fiber collects the collimated signal from the second polarizer and delivers the background-free Brillouin light signal to the single-stage VIPA spectrometer module.

## Confocal Brillouin microscope

A single-longitudinal-mode laser (Cobolt AB, Flamenco) operating at $\lambda = 660$ nm, after entering into an inverted microscope (Olympus, IX73), is focused by a 60x water-immersion objective lens (Olympus UPlanSApo 60x/1.20) to the sample placed on a motorized stage (Fig. 2a). The scattered light is collected by the same objective. A dichroic beamsplitter (DBS) (Di03-R660-t1-25 × 36, Semrock) is then used to register both Raman and Brillouin signals. Raman photons above 670 nm are transmitted by the DBS and then focused by a lens (f = 35 mm, AC254-035-B-ML, ThorLabs) on the entrance slit of a spectrometer (Isoplane160, Princeton instruments) equipped with a grating of 600 gr/mm and connected to a front illuminated CCD (PIXIS256F, Princeton Instruments). The reflected Brillouin signal passes through a quarter waveplate (QWP) and is reflected by a polarizing beams splitter (PBS) to ensure maximum transmission efficiency. The Brillouin scattered light is then coupled into the polarization-maintaining single-mode fiber, providing confocality and flexible beam delivery to the filter and detection modules. Light collimated at the output of the fiber is transmitted through the filter and coupled back into a second single-mode fiber delivering the background-free Brillouin signal to the spectrometer module. The former consists of a VIPA etalon (LightMachinery) where light is focused by a cylindrical lens. Preliminary investigations on milk solutions and water interfaces were performed with a VIPA etalon of FSR = 30 GHz and 515 ± 20 MHz spectral resolution, while bone tissues were analyzed using a VIPA etalon of FSR = 60 GHz and 859 ± 20 MHz spectral resolution. A lens translates the angular content of the output interference fringes into the spatial domain by a Fourier transform, thus yielding the final Brillouin spectrum for single-frame acquisition by a CCD camera (Retiga R1). Unless otherwise specified, Brillouin spectra were acquired with 40 mW optical power at the sample plane and 100 ms data acquisition time.

## Acquisition and processing of Brillouin and Raman spectral maps

Brillouin and Raman spectral maps of bone tissue sections were sequentially registered using a step of 5 μm, covering an area of 645 × 740 μm, acquired by applying an exposure time of 300 ms/pixel (Brillouin) and 500 ms/pixel (Raman) with a laser excitation power of 60 mW at the sample plane. Raw Brillouin spectral frames were processed by retrieving the spectral intensity profiles along the y-axis and performing a least-squared fitting using Lorentzian functions on custom Matlab codes. The spectral axis was calibrated using the spectral reference of distilled water acquired before and after Brillouin image acquisition. Raman data were first calibrated at single spectrum level by performing wave-number axis calibration using toluene and a ArHg lamp, and intensity calibration using a calibrated halogen lamp. Raman spectral maps were then pre-processed using the RamApp toolbox[60]. Cosmic rays were corrected using Modified Z-score. Baseline correction was implemented to decrease fluorescent emissions that could interfere with the investigation of Raman spectra. Spectral denoising was performed by adaptive smoothness penalized least squares (as-PLS) approach based on a Whittaker algorithm. Finally, multivariate analysis through N-FINDR algorithm[61] was performed to select pixels containing apatite, further used as spectral mask for data processing of Brillouin data on the sole pixels containing mineralized bone.

## Test beads preparation

Aqueous suspensions of polystyrene (PS) microparticles (72986-5ML-F, Merck KGaA, Darmstadt, Germany) and of poly(methacrylic acid) (PMMA) microparticles (90515-5ML-F, Merck KGaA, Darmstadt, Germany), of ~8 μm and ~10 μm diameter, respectively, were diluted 100 times and mixed 1:1 in water (1 mL). 2 μl of this suspension were deposited onto fused silica coverslips and air dried for 30 min.

## Animals and sample preparation

The generation of Rankl[+/−] mice has been previously described[44]. The colony was maintained in heterozygosis in the specific pathogen-free facility of Humanitas Research Hospital; the litters were genotyped as described in ref. 44. Three-week-old Rankl[−/−] (Rankl knockout) mice and wildtype littermates were euthanized by $CO_2$ asphyxiation; tissues were harvested and fixed in 4% paraformaldehyde (PFA). For each mouse, part of the lumbar spine was processed for embedding in methylmetacrilate immediately after fixation, without decalcification. Six μm thick sections were laid on polylisine-coated silica fused slides, deplasticized and analyzed. All mouse experimental procedures were performed in accordance with the Humanitas Institutional Animal Care and Use Committee and with international laws (authorization n.11/2019-PR).

## Data availability

The data that support the findings of this study are available from the corresponding authors on request.

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

## Acknowledgements

The authors thank the technical assistance of the CNR senior technician Dario Strina. We further thank Dr Gianluca Galzerano for the help in the spectral data acquisition with the FTIR spectrometer. The authors acknowledge financial support by the European Union's NextGenerationEU Program with the I-PHOQS Infrastructure (IR0000016, ID D2B8D520, CUP B53C22001750006) "Integrated infrastructure initiative in Photonic and Quantum Sciences". The authors further wish to thank the TROPHY (ulTRafast hOlograPHic FTIR microscopY) project funded by the European and Innovation Council under the Horizon Europe program in the PATHFINDEROPEN-01 call (grant N. 101047137).

## Author contributions

G.A. and G.C. conceived the system. C.M. and D.P. contributed in the implementation of the optical filter. R.V. built the Raman setup, supervised Raman data acquisition and analysis, and contributed to biological assessment of data. M.L.S. and C.S. contributed to the bone tissue samples used for the osteopetrosis studies. M.B. contributed to the implementation of the closed-loop system. M.V. performed Raman data collection and processing. All authors contributed to the data interpretation and manuscript writing.

## Competing interests

G.A. and D.P. own shares in the company Specto Srl. The remaining authors declare no competing interests.
