## [Peer Review File · Nature Communications]

Birefringence-induced phase delay enables Brillouin mechanical imaging in turbid mediaEditorial Note: This manuscript has been previously reviewed at another journal that is not operating a transparent peer review scheme. This document only contains reviewer comments and rebuttal letters for versions considered at *Nature Communications*.

REVIEWER COMMENTS

Reviewer #2 (Remarks to the Author):

This manuscript has seen considerable improvement from its previous version, successfully addressing many critiques from three reviewers. Extinction levels of >60 dB have been achieved through other methods, such as three-stage VIPAs, multiple-pass etalons, and gas absorption filters. This work introduces a birefringence filter as a more straightforward alternative, particularly advantageous for Brillouin imaging in turbid samples at depth, as evidenced by the pilot study. Yet, further validation is essential to clarify the system's broad applicability across different sample types.

Regarding temperature sensitivity, an inherent issue with the long crystal filter, the introduction of a tunable liquid crystal filter as a stabilization measure is a significant enhancement. This crucial feature warrants more detailed explanation within the manuscript and figures. In Supplementary Fig. 6, the temperature variation range during experiments should be clarified. Additionally, it would be beneficial to present plot (b) in terms of extinction (dB) and detail the dynamic range of the liquid crystal, including its effectiveness across varying ambient temperatures, from 15 to 30 degrees Celsius.

Wavelength sensitivity: To achieve a 60 dB extinction, the phase delay must be kept within a 0.001 radian error. Given $L = 35$ mm, $\lambda = 660$ nm, and $\Delta_n = 0.23$, the birefringence phase in the long crystal is approximately 77,000 radians. For 0.001 radian precision, λ 's control must be within 1.3×10^{-8} , leading to a stringent requirement of $\Delta_\lambda = 0.0086$ pm. Considering typical laser specifications are < 1 pm (over $\pm 2^\circ\text{C}$ for 8 hours), this sensitivity poses a significant challenge and requires further consideration.

Sample dependence: The effectiveness of the birefringence filter hinges on a specific input polarization state, highlighted by the use of a polarizer in front of the crystal. The performance in scenarios where scattered light exhibits multiple polarization states, such as in birefringent samples like stratified muscles, needs clarification and experimental verification. In such cases, the probe beam's polarization could alter within the sample, resulting in Brillouin scattered light of the opposite circular polarization state, which would then be blocked by the filter's input polarizer. This issue warrants attention.

The term "Delayed Brillouin light scattering" in the title may cause confusion, as it could be interpreted as referring to scattering events that are temporally delayed within a sample, which is misleading. Since all interference filters function based on phase delays, a more precise description would be beneficial.

Similarly, the use of "Delayed Brillouin signal" after Eq. (1) lacks clarity. If the intention is to describe the differential group delay between two birefringence axes, this should be explicitly stated for better understanding.

The abstract's reference to the method being "mostly applied to semi-transparent samples" may be misleading. Biological tissues, including the sclera, are not generally considered semi-transparent, and Brillouin spectroscopy has been applied to opaque materials like bones and cartilage.

Reviewer #3 (Remarks to the Author):

The present version of the paper "Delayed Brillouin light scattering enables all-optical mechanical imaging in turbid media" by G. Antonacci et al. has clarified key points and corrected some of the presented results. Specifically, the introduction of the full-wave liquid crystal retarder that can be finely tuned by applying a controlled voltage has stabilized the performance of the instrument. In the previous version, the temperature fluctuations of the environment strongly affected the system

stability. This effect, evident by the data reported in the new figure 6 of Supplementary Information, appears to be corrected.

Furthermore, in the bones analysis, the use of a different VIPA with an adequate Free Spectral Range (FSR) prevented confusion regarding the overlap of different interference orders, leading to different conclusions about the mechanical properties of the investigated samples.

As I mentioned in my initial review, the paper explores the study of turbid media and materials in proximity to reflective surfaces. While these conditions were already accessible with existing setups, the proposed experimental arrangement offers a compact solution to achieve a contrast of 90 dB using a single VIPA spectrometer. The overall quality of the paper has improved compared to the previous version. However, there are still aspects in the presentation of data and the experimental system that require correction or integration:

1) Pag 10, in order to comprehensively describe the performance of the presented system, it is advisable to include not only the sensitivity but also the spectral resolution of the setup. Additionally, the reported sensitivity of 13 MHz appears very high. It is not directly comparable with the errors indicated in the data, particularly as it was obtained without the implementation of a closed-loop control for tuning the voltage applied to the liquid crystal retarder. Considering the significant impact of environmental thermal fluctuations on system performance, it is recommended to present the sensitivity on the same time scale (1 hour) used to assess the system stability (see Supplementary Figure 6).

2) The Signal-to-Noise Ratio (SNR) presented in the new Figure 3 of the Supplementary Information actually represents the Signal-to-Background Ratio (SBR). The noise of the Brillouin peak should be evaluated by considering the intensity variation of the Brillouin peak. Furthermore, also in this context, it is important to show the evolution as a function of time.

3) The authors claim that the transmission envelope of the BIPD filter does not impact either the frequency shift or the spectral shape of the detected Brillouin peaks. The values of the Brillouin peak of a representative sample are depicted in the new Supplementary Figure 5 ($\nu_{ASB} = 11.57 \pm 0.05$ GHz and $\nu_{ASB} = 11.55 \pm 0.05$ GHz - with a fitting error of 1.3%). For clarity, it is recommended to specify the values for both peaks (Stokes and Anti-Stokes), including the associated peak widths.

4) In Figure 4d), the Full Width at Half Maximum (FWHM) of the Point Spread function (PSF) obtained as the first derivative of the Edge Spread function (ESF) determines the axial resolution of the Brillouin system. However, both curves presented in Figure 4d) exhibit a FWHM larger than the reported value (3.9 ± 0.2) μm . It is reasonable to associate this latter value with the Half Width at Half Maximum (HWHM) of the PSF. Consequently, the resulting spatial resolution is twice the reported value.

5) The reported estimation of the Optical Density (OD) at 350 μm in pure milk is stated to be $OD=7$. Please provide details on how this value was determined. Additionally, the absence of the well-established shoulders arising from the multiple scattering (already observed in milk, latex suspensions, PNIPAM solutions, etc.) needs to be addressed and explained in the text

6) The sentence on page 17 reads, "Remarkably, this represents more than two-order of magnitude decrease in the data acquisition time with respect to previously reported studies on similar samples involving a 6-pass scanning Fabry-Perot interferometer as the detection unit [44-45]" To ensure a fair comparison between the performance of two different setups, it is important to compare the quality of the data using the same sample and the same power density. In the present case, the samples are different and the quality of the data is not considered. Moreover, the presented data have been acquired using a laser power of 60 mW on the samples and using a high numerical aperture water immersion objective (NA=1.2). In refs 44-45, the power was 7 mW and the measurements were performed using a long working distance objective NA=0.42. To accurately report the system's performance, it is important that the authors objectively compare their results with those of other

studies.

An other suggestion:

-in the last years, several studies have been published regarding the spatial resolution of Brillouin microscopy. These papers demonstrated that the phonons properties guide the spatial resolution of Brillouin microscopy, which is not only linked to the optical resolution of the used microscope. The authors should revise the following sentence of the introduction, as it can be misleading: "To answer this need, Brillouin microscopy has emerged as a purely optical and label-free method to measure viscoelastic properties with diffraction-limited spatial resolution in the volume of biological organisms".

Response to Reviewers' Comments

Reviewer #2 (Remarks to the Author):

This manuscript has seen considerable improvement from its previous version, successfully addressing many critiques from three reviewers. Extinction levels of >60 dB have been achieved through other methods, such as three-stage VIPAs, multiple-pass etalons, and gas absorption filters. This work introduces a birefringence filter as a more straightforward alternative, particularly advantageous for Brillouin imaging in turbid samples at depth, as evidenced by the pilot study. Yet, further validation is essential to clarify the system's broad applicability across different sample types.

We thank the reviewer for the positive feedback on our manuscript. Indeed, our filter represents a more compact solution compared to three-stage VIPAs while giving significant advantages compared to gas absorption filters that require non-trivial laser lock-in at specific wavelengths and suffer from multiple absorption lines. We hope that this new version of the manuscript has addressed his/her concerns.

Regarding temperature sensitivity, an inherent issue with the long crystal filter, the introduction of a tunable liquid crystal filter as a stabilization measure is a significant enhancement. This crucial feature warrants more detailed explanation within the manuscript and figures. In Supplementary Fig. 6, the temperature variation range during experiments should be clarified. Additionally, it would be beneficial to present plot (b) in terms of extinction (dB) and detail the dynamic range of the liquid crystal, including its effectiveness across varying ambient temperatures, from 15 to 30 degrees Celsius.

We understand the reviewer request for more information on the liquid crystal retarder as a key element to rapidly tune the phase retardance in our BIPD filter.

Actions taken - To address the reviewer remarks, we have expanded the *BIPD filter module* section in Methods as follows: *A full-wave phase retarder (Thorlabs, LCC1613-A) is used to fine tune the phase delay at the output of the YVO₄ crystal. The phase retarder employs a nematic liquid crystal cell acting as a variable wave plate, allowing active control over the phase delay of the incoming beam by an applied voltage. Specifically, the liquid crystal cell can induce a variable phase retardation within the range of (0-1.2) λ with a typical response time of < 30 ms at -20 to 40 °C nominal operating temperature. As a result, this active element is needed to rapidly set the Rayleigh scattering to be linearly polarized along the orthogonal direction with respect to the transmission axis of the output analyzer, thus correcting for possible laser drifts or thermal expansion of the crystal.*

Using a digital temperature analyzer, we recorded the lab temperature near the filter during standard working hours, i.e. between 6AM and 6PM. Results are shown in Figure R1 (added as the new Supplementary Figure 7).

Figure R1 – Environmental lab temperature variation over 12-hour interval. Data has been recorded every second using a digital temperature analyzer placed in proximity to the filter. Results show a maximum temperature variation of up to 0.5°C over the time of investigation.

We have also added in the text: To compensate for extra phase delays induced by unavoidable environmental temperature fluctuations (Supplementary Fig. 7).

As detailed in the figure caption, Supplementary Fig. 6b shows a plot of the relative Rayleigh intensities measured by the CCD camera during a 1-hour Brillouin spectral acquisition in water. Unfortunately, conversion of such values in dB units would not be representative of the extinction of the filter. In fact, a precise measure of the extinction over time would require continuous reading of the incident optical power to a sample (preferably a highly reflective mirror) and the residual Rayleigh signal at suppression, the former also requiring the use of calibrated neutral density filters to overcome the limited dynamic range of the CCD camera. As such, we believe that the intensity axis in arbitrary units would be more appropriate to demonstrate the effectiveness of the closed loop in real Brillouin spectroscopy acquisitions.

Wavelength sensitivity: To achieve a 60 dB extinction, the phase delay must be kept within a 0.001 radian error. Given $L = 35$ mm, $\lambda = 660$ nm, and $\Delta n = 0.23$, the birefringence phase in the long crystal is approximately 77,000 radians. For 0.001 radian precision, λ 's control must be within 1.3×10^{-8} , leading to a stringent requirement of $\Delta \lambda = 0.0086$ pm. Considering typical laser specifications are < 1 pm (over $\pm 2^\circ\text{C}$ for 8 hours), this sensitivity poses a significant challenge and requires further consideration.

The reviewer raises an important topic. Indeed, in addition to the thermal expansion of the birefringent crystal, a major source of system instability is also given by the thermal frequency drift of the laser. In more detail, a laser stability of < 10 MHz is needed to maintain a 60 dB extinction over time, as the reviewer has correctly noted. Such stability is not guaranteed by commercial lasers, thus imposing the need of active stabilization. Unlike commonly used gas absorption filters that require sophisticated laser lock-in modules, our BIPD filter does not require additional modules as it can rapidly compensate for laser frequency drifts by finely adjusting the phase retardation of the liquid crystal. In other words, the transmission minimum of the filter transfer function can be conveniently moved in the spectral domain with a typical response time of < 50 ms to compensate for both frequency drifts of the laser as well as the thermal expansion of the crystal.

Actions taken – To gain a deeper understanding of our laser stability beyond the nominal value (< 1 pm over $\pm 2^\circ\text{C}$ for 8 hours) provided in the datasheet of the laser manufacturer (Cobolt), we

have performed measurements over a time period of 10 hours co-registering the spectral intensity of the VIPA and the temperature in the lab during the same time interval. Results (Figure R2 and new Supplementary Figure 8 of the manuscript) indicate a laser drift of 320 ± 30 MHz for a temperature change of 0.1°C , below the expected performance indicated by the laser manufacturer. On the other hand, we estimated a frequency drift of 4.8 ± 0.5 MHz per minute, confirming the rate at which we performed the filter re-calibration using the implemented active closed-loop during image acquisition.

Figure R2 – Laser frequency drift. Monochromatic light from the laser was sent directly to the VIPA spectrometer and the resulting spectral profile was recorded for 10 hours. Values for the laser spectral drift (blue left y-axis) were obtained fitting the non-saturated Rayleigh peak for two consecutive interference orders of the VIPA. In parallel, the lab temperature was co-registered using a digital thermometer (red right y-axis). Results show a clear dependence of the laser stability on the ambient temperature, with an estimated laser drift of approximately 320 ± 30 MHz for a temperature change of 0.1°C . In the time domain, we estimated a frequency drift of 4.8 ± 0.5 MHz/min, reflecting our need to re-calibrate the BIPD filter with an active closed-loop control every 1-2 mins.

Sample dependence: The effectiveness of the birefringence filter hinges on a specific input polarization state, highlighted by the use of a polarizer in front of the crystal. The performance in scenarios where scattered light exhibits multiple polarization states, such as in birefringent samples like stratified muscles, needs clarification and experimental verification. In such cases, the probe beam's polarization could alter within the sample, resulting in Brillouin scattered light of the opposite circular polarization state, which would then be blocked by the filter's input polarizer. This issue warrants attention.

The reviewer notes another interesting aspect. Indeed, for a given input polarization state there are few samples that produce a different output polarization state as a consequence of their small intrinsic birefringence. While a change in the output polarization would result in an extra optical loss for the Brillouin signal transmitted by our filter, this would not affect the effectiveness of the filter in suppressing the unwanted Rayleigh scattering while maintaining the analyzed Brillouin peaks unvaried in their spectral content. As the reviewer correctly notes, the polarization state entering in our birefringent crystal is ultimately fixed by the input polarizer (PL in Fig.2a) oriented with its transmission axis at 45 degrees with respect to the optical axis of the birefringent crystal, so that the input Rayleigh and Brillouin signals will always have equal input polarization. Further, we shall note that, while the intrinsic birefringence of specific biological samples is typically low (in the order of 10^{-3}), the use of a polarizing beam splitter (PBS) - a widely adopted solution in most Brillouin microscopy schemes - involves the exact same reduction in

throughput as a consequence of the sample birefringence. In our system, the transmission axis of the input polarizer PL is parallel to the one defined by the PBS, so no extra losses are involved in addition to those introduced by the PBS. One way to avoid losses due to sample birefringence would be to use non-polarizing beam splitters, but this would come at the cost of an overall reduced throughput given by the beam splitting ratio.

Actions taken – We reproduced the effect of a sample birefringence by placing a quarter wave plate before the input polarizer of the filter while collecting Brillouin spectra of water (Figure R3 and new Supplementary Figure 6 of the manuscript). The resulting spectra show the expected drop in the peak intensity when the waveplate was rotated by 45 degrees with respect to the input polarization.

Figure R3 – Response to sample birefringence. While measuring the Brillouin spectra of water, a quarter wave plate was placed before the filter input polarizer to mimic the sample birefringence. With an appropriate choice of the wave plate optical axis, the polarization of the Brillouin (B) and Rayleigh (R) signals was turned to circular (red) from the original linear state (blue) parallel to the transmission axis (black) of the filter input polarizer. Apart from the expected extra insertion loss, a change in the input polarization did not affect the spectral shift of the Brillouin peaks ($\nu_B^{lin} = 6.04$ GHz; $\nu_B^{circ} = 6.06$ GHz) nor the ability of the filter to fully suppress the Rayleigh signal.

In the main text we have added the sentence: Moreover, we observed no significant variations in the Brillouin peak shape and frequency shift in response to either a varying phase delay (Supplementary Fig. 5) or a change in the output polarization of the Rayleigh and Brillouin signals that may be caused by small sample birefringence (Supplementary Fig. 6).

The term "Delayed Brillouin light scattering" in the title may cause confusion, as it could be interpreted as referring to scattering events that are temporally delayed within a sample, which is misleading. Since all interference filters function based on phase delays, a more precise description would be beneficial.

We appreciate this point raised by the reviewer. To avoid confusion, the title has now been changed to *Birefringence-induced phase delay enables Brillouin mechanical imaging in turbid media*.

Similarly, the use of "Delayed Brillouin signal" after Eq. (1) lacks clarity. If the intention is to describe the differential group delay between two birefringence axes, this should be explicitly stated for better understanding.

We apologize for the confusion. The sentence has been re-written as follows: *Conversely, the filter can apply to the Brillouin signal a convenient relative phase retardation which sets its polarization to be linear and parallel to the analyzer transmission axis (Fig. 1c).*

The abstract's reference to the method being "mostly applied to semi-transparent samples" may be misleading. Biological tissues, including the sclera, are not generally considered semi-transparent, and Brillouin spectroscopy has been applied to opaque materials like bones and cartilage.

It was indeed not the authors' intention to state that Brillouin microscopy has only been applied to semi-transparent samples. Indeed, previous studies on bones are correctly referenced in the main text. On the other hand, it is in our opinion true that the majority of the applications demonstrated so far involved semi-transparent samples.

Actions taken – To avoid confusion, we have replaced the sentence in the abstract with the following: *Yet, the effectiveness of this method is hindered when applied to turbid samples as a consequence of the dominant elastic background light that overwhelms the Brillouin peaks.*

Reviewer #3 (Remarks to the Author):

The present version of the paper "Delayed Brillouin light scattering enables all-optical mechanical imaging in turbid media" by G. Antonacci et al. has clarified key points and corrected some of the presented results. Specifically, the introduction of the full-wave liquid crystal retarder that can be finely tuned by applying a controlled voltage has stabilized the performance of the instrument. In the previous version, the temperature fluctuations of the environment strongly affected the system stability. This effect, evident by the data reported in the new figure 6 of Supplementary Information, appears to be corrected.

Furthermore, in the bones analysis, the use of a different VIPA with an adequate Free Spectral Range (FSR) prevented confusion regarding the overlap of different interference orders, leading to different conclusions about the mechanical properties of the investigated samples.

We appreciate that the reviewer acknowledges the important improvement in the quality of our manuscript following the previous review round. We hope that the new revised version of the manuscript can fully address his/her remaining concerns.

As I mentioned in my initial review, the paper explores the study of turbid media and materials in proximity to reflective surfaces. While these conditions were already accessible with existing setups, the proposed experimental arrangement offers a compact solution to achieve a contrast of 90 dB using a single VIPA spectrometer. The overall quality of the paper has improved compared to the previous version.

However, there are still aspects in the presentation of data and the experimental system that require correction or integration:

We thank the reviewer for recognizing the advance in Brillouin microscopy allowed by our birefringent filter and the overall improved quality of our manuscript. In the following, we address the remaining concerns.

Pag 10, in order to comprehensively describe the performance of the presented system, it is advisable to include not only the sensitivity but also the spectral resolution of the setup.

We agree that the information on spectral resolution gives a more comprehensive understanding of our system performance.

Actions taken – In page 10 we have added the following specification: *The microscope included a single-stage, single-pass VIPA spectrometer module with measured spectral contrast of 30 ± 3 dB*

(Supplementary Fig. 2), spectral resolution of 515 ± 20 MHz and sensitivity of 14.5 ± 2 MHz (Supplementary Fig. 3).

In Supplementary Figure 2, we have added the following statement in the figure caption: *Fitting the non-saturated elastic peaks with a Lorentzian function, we measured a spectral resolution of $\text{FWHM}=(515\pm 20)$ MHz.*

In the *Confocal Brillouin microscope* section of Methods we have further added: *Preliminary investigations on milk solutions and water interfaces were performed with a VIPA etalon of FSR 30 GHz and 515 ± 20 MHz spectral resolution, while bone tissues were analyzed using a VIPA etalon of FSR 60 GHz and 859 ± 20 MHz spectral resolution.*

Additionally, the reported sensitivity of 13 MHz appears very high. It is not directly comparable with the errors indicated in the data, particularly as it was obtained without the implementation of a closed-loop control for tuning the voltage applied to the liquid crystal retarder. Considering the significant impact of environmental thermal fluctuations on system performance, it is recommended to present the sensitivity on the same time scale (1 hour) used to assess the system stability (see Supplementary Figure 6).

We understand the reviewer request to provide a value of sensitivity that reflects an investigation over a longer time scale that includes the operation of the closed loop.

Actions taken – We have updated Supplementary Figure 3a (below as Figure R4) with the extended dataset over 1 hour of spectral acquisition with the active closed loop. The sensitivity obtained from this larger dataset is 14.5 MHz, not significantly different compared to the previously reported figure. Both figure caption and main text have been updated with the new result.

Figure R4 - Sensitivity of the single-stage VIPA spectrometer. The Brillouin spectrum of distilled water was acquired over 60 minutes with a pixel dwell time of 100 ms and fitted using a Lorentzian function. The instrumental sensitivity defined by the standard deviation of the measured frequency shifts is 14.5 MHz.

The Signal-to-Noise Ratio (SNR) presented in the new Figure 3 of the Supplementary Information actually represents the Signal-to-Background Ratio (SBR). The noise of the Brillouin peak should be evaluated by considering the intensity variation of the Brillouin peak. Furthermore, also in this context, it is important to show the evolution as a function of time.

We thank the reviewer for the observation. In our manuscript we defined the Signal-to-Background as the ratio between the Brillouin peak intensity and the average background level taken between the Brillouin and the Rayleigh peaks. This gave a measure of the effective background suppression over time as shown in Supplementary Figure 6d. On the other hand, we do agree with the reviewer that the reported values in Supplementary Figure 3b were not representative of the true SNR.

Actions taken – We have updated Supplementary Figure 3b (below as Figure R5) with the measured Brillouin peak intensity over 60 mins. We calculated the SNR as the average of the reported intensity values over the standard deviation of the same dataset.

Figure R5 - *Signal-to-noise ratio (SNR) characterization. A $SNR=28.6\pm0.5$ was obtained dividing the average Brillouin peak intensity by the standard deviation of the data points collected over the same time period.*

The authors claim that the transmission envelope of the BIPD filter does not impact either the frequency shift or the spectral shape of the detected Brillouin peaks. The values of the Brillouin peak of a representative sample are depicted in the new Supplementary Figure 5 ($\nu_{ASB} = 11.57 \pm 0.05$ GHz and $\nu_{SB} = 11.55 \pm 0.05$ GHz - with a fitting error of 1.3%). For clarity, it is recommended to specify the values for both peaks (Stokes and Anti-Stokes), including the associated peak widths.

We agree with the reviewer that adding the values also for the Stokes peak together with the associated linewidth can provide a more comprehensive understanding of the impact of the filter transmission of the Brillouin spectrum.

Actions taken – The caption of Supplementary Figure 5 has been updated as follows: *Brillouin spectra of a polystyrene test sample acquired with the filter transmission function (dotted line) centred between the Stokes and anti-Stokes Brillouin peaks (red) and at the frequency of the anti-Stokes Brillouin peak (black). In the first case, the frequency shift and linewidth of the anti-Stokes Brillouin peak were measured by Lorentzian fitting to be $\nu_{ASB}=11.33\pm0.05$ GHz and $\Delta\nu_{ASB}=1.01\pm0.10$ GHz respectively, while for the Stokes Brillouin peak we obtained $\nu_{SB}=-11.33\pm0.05$ GHz and $\Delta\nu_{SB}=1.16\pm0.10$ GHz (fit error 1.4%, $R>0.99$). Similarly, the measured frequency shift and linewidth after translation of the filter transmission function were $\nu_{ASB}=11.35\pm0.05$ GHz and $\Delta\nu_{ASB}=1.02\pm0.10$ GHz for the anti-Stokes Brillouin peak, and $\nu_{SB}=-11.30\pm0.05$ GHz and $\Delta\nu_{SB}=0.98\pm0.10$ GHz for the Stokes Brillouin peak (fit error 1.8%, $R>0.99$). Results demonstrate that the Brillouin spectrum is not affected by the transmission envelope of the BIPD filter.*

In Figure 4d), the Full Width at Half Maximum (FWHM) of the Point Spread function (PSF) obtained as the first derivative of the Edge Spread function (ESF) determines the axial resolution of the Brillouin system. However, both curves presented in Figure 4d) exhibit a FWHM larger than the reported value (3.9 ± 0.2) μm . It is reasonable to associate this latter value with the Half Width at Half Maximum (HWHM) of the PSF. Consequently, the resulting spatial resolution is twice the reported value.

Indeed, we previously indicated the resolution as the Half Width at Half Maximum (HWHM) of the retrieved PSF. We have now corrected the value with the Full Width at Half Maximum (FWHM), i.e. 7.8 ± 0.2 μm .

5) The reported estimation of the Optical Density (OD) at 350 μm in pure milk is stated to be $\text{OD}=7$. Please provide details on how this value was determined. Additionally, the absence of the well-established shoulders arising from the multiple scattering (already observed in milk, latex suspensions, PNIPAM solutions, etc.) needs to be addressed and explained in the text

We estimated the optical density using the analytical expression $\text{OD} = l \cdot \mu_e$, where μ_e is the sum of the scattering and absorption coefficients being 20 mm^{-1} (see E. Berrocal, et al. "Laser light scattering in turbid media Part I: Experimental and simulated results for the spatial intensity distribution," *Opt. Express* 15, 10649-10665 (2007)).

The reviewer further raises an interesting aspect regarding the absence of the shoulders from multiple scattering (MS). We attribute the absence of the shoulders to the fact that the MS light contribution is fundamentally depolarized, thus being significantly attenuated when passing through the input polarizer of the filter. However, further investigation will need to be done in order to better characterize this interesting aspect.

Actions taken – We have added the following description in the caption of Supplementary Figure 10: *Values of the Optical Density (OD) were retrieved from the relationship $\text{OD} = z \cdot \mu_e$, where z is the penetration depth and μ_e is the sum of the scattering and absorption coefficients. In the present study, we assumed $\mu_e = 20 \text{ mm}^{-1}$ [1,2].*

We have added a new *Supplementary Reference* section with the following references:
[1] E. Berrocal, et al., *Laser light scattering in turbid media Part I: Experimental and simulated results for the spatial intensity distribution*, *Opt. Express* 15, 10649-10665 (2007)
[2] M.D. Waterworth, et al. *Optical transmission properties of homogenised milk used as a phantom material in visible wavelength imaging*, *Australas Phys Eng Sci Med.*;18(1), 39-44 (1995)

Moreover, we have added the following comment to the main text: *The reason for this absence may be linked to the fact that the MS light contribution is fundamentally depolarized so that it is partially removed when passing through the input polarizer of the BIPD filter.*

6) The sentence on page 17 reads, "Remarkably, this represents more than two-order of magnitude decrease in the data acquisition time with respect to previously reported studies on similar samples involving a 6-pass scanning Fabry-Perot interferometer as the detection unit [44-45]" To ensure a fair comparison between the performance of two different setups, it is important to compare the quality of the data using the same sample and the same power density. In the present case, the samples are different and the quality of the data is not considered. Moreover, the presented data have been acquired using a laser power of 60 mW on the samples and using a high numerical aperture water immersion objective (NA=1.2). In refs 44-45, the power was 7 mW and the measurements were performed using a long working distance objective NA=0.42. To accurately report the system's performance, it is important that the authors objectively compare their results with those of other studies.

We agree with the reviewer on the fact that it is not trivial to provide a direct comparison on the speed between the two different techniques as parameters such as optical power density, objective lenses and the samples themselves may affect the resulting data acquisition time. We further note that besides the parameters correctly indicated by the reviewer, we should also consider the different laser wavelength (660nm) used in this study, which returns a Brillouin signal weaker by a factor 2.4 compared to a 532nm wavelength used in previous studies as a consequence of the λ^{-4} dependence of the signal. In light of these considerations, we believe that our detection system involving the BIPD filter and a single-stage VIPA spectrometer still provides a considerable speed enhancement with respect to other Brillouin imaging systems based on the high-contrast tandem and multipass Fabry-Perot interferometers previously used to study bone.

Actions taken – To address the reviewer’s concern, we have rephrased the sentence as follows: *Remarkably, this represents ~~more than two order of magnitude~~ a significant decrease in the data acquisition time with respect to previously reported studies on similar samples involving a 6-pass scanning Fabry-Perot interferometer as the detection unit [44-45]*”

Another suggestion:

-in the last years, several studies have been published regarding the spatial resolution of Brillouin microscopy. These papers demonstrated that the phonons properties guide the spatial resolution of Brillouin microscopy, which is not only linked to the optical resolution of the used microscope.

The authors should revise the following sentence of the introduction, as it can be misleading:

“To answer this need, Brillouin microscopy has emerged as a purely optical and label-free method to measure viscoelastic properties with diffraction-limited spatial resolution in the volume of biological organisms”. Summary of the key results: The submitted manuscript proposed a birefringent filter as an alternative strategy to reduce the elastic scattering background in Brillouin microscopy.

We thank the reviewer for the correct observation.

Actions taken – We have replaced the term *diffraction-limited* with *subcellular*. Moreover, we have added the following sentence in the introduction: *While spectral broadening is minimized in such configuration [Antonacci, G. et al. Applied Physics Letter (2013)], the ultimate spatial resolution of the technique must also take into account the extension of the probed acoustic phonons that may be larger than the system Point Spread Function (PSF) [Caponi, S. et al. Optics Letter (2022)].*

REVIEWER COMMENTS

Reviewer #2 (Remarks to the Author):

The authors have adequately addressed most of the concerns, with one notable exception related to clarity. The integration of the variable birefringence filter is pivotal for ensuring stability against temperature changes and laser frequency drift, as well as for correcting small birefringence errors in the passive birefringence filter. The critical component warrants a clear and early mention in both the Abstract and main text, as well as depiction in the schematic diagrams of the principle and setup in Figures 1 and 2.

Furthermore, the strength of this research lies in its performance. For this, the tuning element is essential, as evident in the additional supplementary data. It is recommended some of the data be integrated into the main manuscript figures to highlight the indispensable role of feedback tuning.

Since a few alternative filter designs have achieved 65 dB extinction, the use of the term "unparalleled" in the Abstract seems overreaching. It would be advisable to either select a different term or rephrase the sentence to provide a more accurate context.

Reviewer #3 (Remarks to the Author):

The present version of the paper "Birefringence-induced phase delay enables Brillouin mechanical imaging in turbid media" by G. Antonacci et al. has clarified the key points reported in my previous review and has integrated the data accordingly.

However, I believe that the explanation for the absence of shoulders in the Brillouin peak due to multiple scattering needs to be rephrased. The authors suggest:

"The reason for this absence may be linked to the fact that the MS light contribution is fundamentally depolarized so that it is partially removed when passing through the input polarizer of the BIPD filter"

In the multiple scattering regime, achieved at about $OD=10$, the MS light becomes randomly polarized compared to the incident one. Consequently, we can imagine that the MS light presents the two polarization components with equal probability. In the Brillouin spectrum of opaque samples, the depolarized component is principally due to the MS, so the input polarizer of the BIPD filter eliminates half of the MS light intensity, thereby reducing the presence of the expected shoulders.

Typos:

-Pag. 3: in the definition of the scattering angle, K_{in} and K_s are unit vectors, otherwise the definition should be corrected by normalizing the product

-pag 5 the spectral resolution of TFP is (~ 100 MHz)

Response to Reviewers' Comments

Reviewer #2 (Remarks to the Author):

The authors have adequately addressed most of the concerns, with one notable exception related to clarity. The integration of the variable birefringence filter is pivotal for ensuring stability against temperature changes and laser frequency drift, as well as for correcting small birefringence errors in the passive birefringence filter. The critical component warrants a clear and early mention in both the Abstract and main text, as well as depiction in the schematic diagrams of the principle and setup in Figures 1 and 2.

We thank the Reviewer for the positive feedback on our revised version of the manuscript. We have addressed all his/her remaining remarks, which we believe have further improved the quality and clarity of our work.

Action taken: To address the Reviewer remark, we have modified the abstract as follows: *The filter exploits the phase delay induced by a birefringent crystal and finely tuned by a liquid crystal retarder to modify the relative polarization states of the Brillouin and Rayleigh signals, enabling the rejection of the background light using a polarizer.*

Moreover, we have included the presence of the active liquid crystal retarder in the schematic of Figure 1 (Figure R1) and in the caption:

Figure R1: (...) **b** Phase delay along the birefringent media. The anisotropy of the birefringent crystal induces a phase delay so that after a specific length L the output signals exhibit a relative phase retardation $\Delta\phi = \pi$. A full-wave liquid crystal (LC) retarder ensures linear and orthogonal polarization states for the Rayleigh and Brillouin signals by fine tuning the phase delay of the beam transmitted by the birefringent crystal. (...)

In the main text we have modified and added the following sentences:

For a given material birefringence Δn , it is therefore possible to select a crystal thickness such that the relative phase variation $\Delta\phi = \pi$. the resulting output Brillouin and Rayleigh polarization states are linear and orthogonal with respect to each other.

(...) In addition, a fast full-wave liquid crystal (LC) retarder enables active fine-tuning of the phase retardation of the beams transmitted by the birefringent crystal to guarantee a linear polarization state for the Rayleigh signal. As a result, the Rayleigh signal is fully rejected by a linear polarizer (analyzer) with transmission axis oriented orthogonal to its polarization. Conversely, the Brillouin signal with polarization orthogonal with respect to the Rayleigh signal and parallel to the analyzer transmission axis is transmitted (Fig. 1c).

We note that Figure 2 (Figure R2) already includes the presence of the LC element in the BIPD schematics and caption as follows: *The full-wave LC retarder actively tunes the phase delay of the beam transmitted by the YVO₄ crystal, ensuring a perpendicular linear polarization for the Rayleigh signal with respect to the transmission axis of the analyzer (AL).*

Furthermore, the strength of this research lies in its performance. For this, the tuning element is essential, as evident in the additional supplementary data. It is recommended some of the data be integrated into the main manuscript figures to highlight the indispensable role of feedback tuning.

We acknowledge the Reviewer feedback and modify the manuscript accordingly.

Action taken: We have integrated former Supplementary Fig. 9a as a new panel c of Figure 2 (below as Figure R2).

Figure R2

In the Figure caption we have modified the following sentence: *By applying a voltage to the LC retarder (c), the filter can be tuned to suppress the Rayleigh peak, making the Brillouin peaks clearly visible (d).*

In the main text, we have modified the sentence accordingly: *By actively tuning the LC retarder (Fig. 2c), we aligned the transmission minimum of the filter to the laser wavelength. In turn, the Rayleigh peak was completely suppressed, making the Brillouin peaks clearly visible (Fig. 2d and Supplementary Video 1).*

Supplementary Fig. 9 (below as Figure R3) and relative caption have been also updated accordingly.

Figure R3: Closed-loop control for filter stabilization. A direct comparison of the filter performance in terms of relative Rayleigh intensity (a), frequency shift (b) and signal-to-background (c) with and without feedback loop shows a significant enhancement of the filter stability over 60 mins. To align the filter at the laser wavelength, the liquid crystal was tuned by voltage sweep at 20 Hz with an incremental step of 1 mV, corresponding to a phase delay of ~ 0.5 mrad, and the transmitted Rayleigh intensity was measured sequentially using the CCD camera to determine the optimal phase retardance of the liquid crystal. This filter calibration process was iterated every 1 min resulting in almost negligible ($\sim 3\%$) extra contribution in the overall data acquisition time.

Since a few alternative filter designs have achieved 65 dB extinction, the use of the term “unparalleled” in the Abstract seems overreaching. It would be advisable to either select a different term or rephrase the sentence to provide a more accurate context.

We agree that there are other optical schemes that can reach 65 dB extinction; such systems require cascaded elements (e.g. multistage VIPAs) or multipass schemes (e.g. tandem Fabry-Perot interferometers). However, to the best of our knowledge, no filters have been demonstrated to reach 65 dB extinction using only single-pass and single-stage configuration, as introduced here.

Action taken: To make the comparison more specific, we have rephrased the sentence in the abstract as follows: *Here, we introduce a common-path Birefringence-Induced Phase Delay (BIPD) filter exhibiting an extinction ratio of 65 dB in a single-pass configuration.*

Reviewer #3 (Remarks to the Author):

The present version of the paper “Birefringence-induced phase delay enables Brillouin mechanical imaging in turbid media” by G. Antonacci et al. has clarified the key points reported in my previous review and has integrated the data accordingly.

We thank the Reviewer for the time and effort in further reviewing our manuscript. We hope that this revised version has now addressed all his/her remaining remarks.

However, I believe that the explanation for the absence of shoulders in the Brillouin peak due to multiple scattering needs to be rephrased. The authors suggest: “The reason for this absence may be linked to the fact that the MS light contribution is fundamentally depolarized so that it is partially removed when passing through the input polarizer of the BIPD filter”

In the multiple scattering regime, achieved at about $OD=10$, the MS light becomes randomly polarized compared to the incident one. Consequently, we can imagine that the MS light presents the two polarization components with equal probability. In the Brillouin spectrum of opaque samples, the depolarized component is principally due to the MS, so

the input polarizer of the BIPD filter eliminates half of the MS light intensity, thereby reducing the presence of the expected shoulders.

Action taken: We have rephrased the sentence as follows: *This absence may be linked to the fact that, in the MS regime, light becomes randomly polarized compared to the incident one. As a result, the input polarizer of the BIPD filter rejects half of the MS light signal, thereby reducing the presence of the expected shoulders.*

Typos:

-Pag. 3: in the definition of the scattering angle, K_{in} and K_s are unit vectors, otherwise the definition should be corrected by normalizing the product

Indeed, we used the (^) symbol on top of K_{in} and K_s that conventionally refers to unit vectors. To make this more clear we have also specified it to the main text as follows: "(...) defined by the incident and scattered unit wave vectors".

-pag 5 the spectral resolution of TFP is (~ 100 MHz)

Noted. We have updated the text accordingly.